# Early Changes in LIPI Score Predict Immune-Related Adverse Events: A Propensity Score Matched Analysis in Advanced Non-Small Cell Lung Cancer Patients on Immune Checkpoint Blockade

**DOI:** 10.3390/cancers16020453

**Published:** 2024-01-20

**Authors:** Fabrizio Nelli, Agnese Fabbri, Antonella Virtuoso, Diana Giannarelli, Julio Rodrigo Giron Berrios, Eleonora Marrucci, Cristina Fiore, Enzo Maria Ruggeri

**Affiliations:** 1Medical Oncology Unit, Central Hospital of Belcolle, Department of Oncology and Hematology, Strada Sammartinese snc, 01100 Viterbo, Italyenzo.ruggeri@asl.vt.it (E.M.R.); 2Biostatistics Unit, Scientific Directorate, Fondazione Policlinico Universitario A. Gemelli, Istituto di Ricovero e Cura a Carattere Scientifico (IRCCS), 00168 Rome, Italy; diana.giannarelli@policlinicogemelli.it

**Keywords:** non-small cell lung cancer, immune checkpoint inhibitors, lung immune prognostic index, immune-related adverse events, survival, first-line therapy, second-line therapy

## Abstract

**Simple Summary:**

The lung immune prognostic index (LIPI) is a valuable tool for reflecting the systemic inflammatory status by combining lactate dehydrogenase (LDH) levels and derived neutrophil/lymphocyte ratio (dNLR) values. In advanced non-small cell lung cancer, the LIPI score before starting PD-(L)1 checkpoint inhibitors can predict disease outcomes through three risk categories. Our hypothesis is that changes in the LIPI score during treatment can predict the likelihood of immune-related adverse events (irAEs). Interestingly, patients who experienced a shift towards a more favorable prognostic category in their LIPI score were at a significantly higher risk of developing irAEs. Furthermore, the dynamic variations in LIPI score provided prognostic insights into overall survival outcomes.

**Abstract:**

In advanced cancer patients undergoing immune checkpoint blockade, the burden of immune-related adverse events (irAEs) is high. The need for reliable biomarkers for irAEs remains unfulfilled in this expanding therapeutic field. The lung immune prognostic index (LIPI) is a noninvasive measure of systemic inflammation that has consistently shown a correlation with survival in various cancer types when assessed at baseline. This study sought to determine whether early changes in the LIPI score could discriminate the risk of irAEs and different survival outcomes in advanced non-small cell lung cancer (NSCLC) patients receiving PD-(L)1 blockade-based therapies. We included consecutive patients diagnosed with metastatic NSCLC who received pembrolizumab, nivolumab, or atezolizumab as second-line therapy following platinum-based chemotherapy, or first-line pembrolizumab either alone or in combination with platinum-based chemotherapy. The LIPI score relied on the combined values of derived neutrophil/lymphocyte ratio (dNLR) and lactate dehydrogenase. Their assessment at baseline and after two cycles of treatment allowed us to categorize the population into three subgroups with good (LIPI-0), intermediate (LIPI-1), and poor (LIPI-2) prognosis. Between April 2016 and May 2023, we enrolled a total of 345 eligible patients, 165 (47.8%) and 180 (52.2%) of whom were treated as first- and second-line at our facility, respectively. After applying propensity score matching, we considered 83 relevant patients in each cohort with a homogeneous distribution of all characteristics across the baseline LIPI subgroups. There was a noticeable change in the distribution of LIPI categories due to a significant decrease in dNLR values during treatment. Although no patients shifted to a worse prognosis category, 20 (24.1%) transitioned from LIPI-1 to LIPI-0, and 7 (8.4%) moved from LIPI-2 to LIPI-1 (*p* < 0.001). Throughout a median observation period of 7.3 (IQR 3.9–15.1) months, a total of 158 irAEs (63.5%) were documented, with 121 (48.6%) and 39 (15.7%) patients experiencing mild to moderate and severe adverse events, respectively. Multivariate logistic regression analysis showed that the classification and changes in the LIPI score while on treatment were independent predictors of irAEs. The LIPI-0 group was found to have significantly increased odds of experiencing irAEs. Following a median follow-up period of 21.1 (95% CI 17.9–25.8) months, the multivariable Cox model confirmed LIPI categorization at any given time point as a significant covariate with influence on overall survival, irrespective of the treatment line. These findings suggest that reassessing the LIPI score after two cycles of treatment could help pinpoint patients particularly prone to immune-related toxicities. Those who maintain a good LIPI score or move from the intermediate to good category would be more likely to develop irAEs. The continuous assessment of LIPI provides prognostic insights and could be useful for predicting the benefit of PD-(L)1 checkpoint inhibitors.

## 1. Introduction

The advent of immune checkpoint inhibitors (ICIs) has radically changed the therapeutic landscape of solid tumors [1]. Unprecedented survival benefits achieved in malignancies resistant to conventional therapies have gained approval from worldwide regulatory agencies for several cancer types [2]. The versatility of ICIs relies on their mechanism of action, which differs from that of cytotoxic chemotherapy and targeted therapies [3]. Monoclonal antibodies targeting receptors of immune inhibitory pathways, including cytotoxic T lymphocyte-associated antigen 4 (CTLA-4), programmed death protein 1 (PD-1), and its ligand 1 (PD-L1), work by restoring the response of T lymphocytes and NK cells against tumor cells and reducing their escape from immune surveillance [4]. The downside due to the extensive use of ICIs concerns the occurrence of immune-related adverse events (irAEs) [5]. Although several alternative reasons have been proposed, impaired self-tolerance resulting in an autoimmune phenotype remains the most reliable mechanism underlying the development of irAEs [6]. In this regard, the most common irAEs resemble inflammatory or autoimmune disorders and may vary depending on specific agents or their combination, the duration of treatment and its dosing, and patients’ or tumor characteristics [7]. The clinical burden of irAEs is high, as mild to moderate toxicities are common and can worsen into serious or life-threatening complications for patients given ICIs [8,9]. Evidence that lower grade irAEs have a favorable impact on long-term survival in patients with various advanced malignancies provides further insight into the complex interplay between ICIs and their immune-modulating effects [10,11]. Conversely, higher-grade irAEs are associated with a worse prognosis due to discontinuation of immunotherapy or the need for systemic immunosuppressive therapy [12]. Therefore, predicting which patients are at increased risk of severe immune-related toxicities before or shortly after the initiation of immunotherapy would allow early diagnosis and treatment of irAEs. Although current studies have focused on host factors, immune microenvironment, and immunogenetics, hematologic parameters have emerged as the most convenient predictors for assessing the risk of irAEs [13]. Soluble indices in peripheral blood may reflect the pro-inflammatory status of the host and the systemic immune response to cancer-related inflammation. Among them, the lung immune prognostic index (LIPI), based on derived neutrophil/lymphocyte ratio (dNLR) and lactate dehydrogenase (LDH), has been proposed as a validated prognostic tool in patients with advanced non-small cell lung cancer (NSCLC) receiving ICIs [14,15,16]. Several studies have also investigated the potential of LIPI as a predictive biomarker for the development of irAEs [17,18,19]. In this regard, a combined score that can provide insights into the efficacy and toxicity of immunotherapy would be highly valuable in clinical practice. However, these studies have relied on limited sample sizes and produced conflicting results. Moreover, despite evidence that PD-(L)1 inhibitors can directly influence peripheral white cells [20,21] and LDH levels [22], all these studies focused on the predictive value of LIPI at baseline. The impact of LIPI changes during ICI treatment on the onset of irAEs has not been addressed previously and remains unknown in a real-life setting. Therefore, the current study aimed to evaluate the predictive value of early changes in LIPI on the development of irAEs in a large population of advanced NSCLC patients undergoing PD-(L)1 blockade. We also investigated whether these variations could affect survival outcomes.

## 2. Materials and Methods

### 2.1. Design and Participants

Conducting a retrospective cohort analysis, we examined patients with advanced NSCLC who received PD-(L)1 inhibitors in a real-world setting. The treatments considered were pembrolizumab, nivolumab, or atezolizumab used as second-line therapy after platinum-based chemotherapy, or first-line pembrolizumab used alone or in combination with platinum-based chemotherapy. Our study included consecutive patients diagnosed with stage IV NSCLC and Eastern Cooperative Oncology Group Performance Status (ECOG PS) 0–2 who underwent at least two treatment courses at our facility from April 2016 to May 2023. All anti-PD-(L)1 agents were prescribed in accordance with their official labels. To ensure consistency, we excluded patients with unavailable PD-L1 tumor proportion score (TPS), activating mutations in EGFR, ALK, or ROS1, and those who had received high-dose corticosteroid therapy or other immunosuppressive medication within 14 days prior to treatment initiation. Patients with asymptomatic or stable brain metastases after radiotherapy were eligible to participate. Our study adhered to the Strengthening the Reporting of Observational Studies in Epidemiology (STROBE) guidelines and was approved by the referring Ethics Committee (registration code: Oss-R-281; protocol number: 855/CE Lazio1). Prior to participation, all patients provided written informed consent for the study procedures and the use of deidentified clinical data for research purposes.

### 2.2. Data Collection and Assessments

Patient and tumor characteristics, as well as information on concomitant medications, disease response, and survival outcomes, were gathered from the government agency’s registry that tracks high-cost drug prescriptions [23]. The monitoring agency guidelines require a comprehensive reassessment of clinical conditions as a prerequisite for continued treatment with anti-PD-(L)1 agents. Before each cycle of therapy, all patients underwent a complete physical examination and laboratory tests, including indices of systemic inflammatory response and thyroid, renal, hepatobiliary, pancreatic, adrenocortical, pituitary, and muscle function. The prospective pharmacovigilance database of the same agency provided details on the frequency and severity of irAEs [24]. The attending physician determined the definition and grading of irAEs using the Common Criteria for Toxicity (CTC)-AE version 5.0 [25]. The 22C3 pharmDx anti-PD-L1 antibody and the Dako Omnis platform were used to perform an immunohistochemical assessment of PD-L1 TPS according to the manufacturer’s recommendations (Agilent Technologies, Inc., Santa Clara, CA, USA). The TPS was defined as the percentage of at least 100 viable tumor cells exhibiting positive membrane staining for PD-L1 expression. Laboratory tests, such as complete blood cell counts and LDH levels, were conducted at specific time points, either before treatment commenced, before the third cycle, or within 30 days after the second cycle if treatment was discontinued early. The range of normal variability of LDH was 135 to 225 units per liter (U/L). All of this information was obtained from medical records in our institutional database. To calculate the LIPI score, we used the values of dNLR (neutrophils/[leukocyte-neutrophils]) and LDH at baseline (basal-LIPI) and during treatment (on-LIPI), following previously established cut-off thresholds [14]. The LIPI score classification identifies a dNLR greater than 3 and an LDH level above the upper limit of normal (ULN) as unfavorable risk factors. Their combination allowed the participants to be divided into three subgroups with a good (no risk factors, or LIPI-0), intermediate (one risk factor, or LIPI-1), or poor (two risk factors, or LIPI-2) prognosis. The categorization of LIPI was utilized to analyze the impact on the frequency and severity of irAEs observed in this particular group of patients. Additionally, the effects of changes in LIPI scoring on progression-free survival (PFS) and overall survival (OS) outcomes were assessed as secondary endpoints. According to the recommendations of the monitoring agency, a baseline assessment of disease extent was conducted prior to treatment initiation, with subsequent restaging occurring every 12 to 16 weeks [23]. Radiologic responses were evaluated using RECIST 1.1 criteria [26]. PFS was calculated from the initial administration of an anti-PD-(L)1 agent until disease progression or death, while OS was measured from the first administration of the agent until death. Patients who did not experience progression or death at the time of the last follow-up were censored (cut-off date November 30, 2023).

### 2.3. Statistical Analysis

Clinical data were analyzed by descriptive statistics using a mean with standard deviation (SD) for normally distributed variables and a median with a 95% confidence interval (CI) or interquartile range (IQR) for skewed variables. Patient characteristics were classified according to basal-LIPI subgroups. Comparative assessments were performed by applying Pearson’s *χ*^2^ test for categorical data and the Kruskal–Wallis test for continuous variables, as appropriate. We planned propensity score matching (PSM) in case the distribution of clinical factors showed an imbalance. The LIPI-0 subgroup was designed as a reference category for comparison of patient characteristics with the other two subgroups. We used the PSM method to obtain a balance in baseline clinical and pathological covariates that differed significantly between LIPI-1 and LIPI-0 and between LIPI-2 and LIPI-0, respectively. Propensity scores were calculated using a logistic regression model that included variables imbalanced across LIPI categories. Matching was based on the nearest-neighbor algorithm with a ratio of 1:1, without replacement, and with a caliper width of 0.1. Accordingly, two well-matched 1:1 pairs were derived relying on the corresponding propensity scores of LIPI-1 and LIPI-2 compared with those of the reference LIPI-0 subgroup. Finally, from these PSM-derived cohorts, we extracted overmatched patients to formulate the final PSM dataset, based on an equal number of patients with comparable clinical and pathological characteristics in each LIPI category. We conducted a multivariate analysis of LDH and dNLR by fitting a linear generalized model on their basal and on-treatment values as a function of predefined covariates. Based on a receiver operating characteristic (ROC) curve calculated at the same time points, we evaluated the sensitivity and specificity of LDH levels and dNLR values in predicting the likelihood of irAEs. The association between patient characteristics, clinical factors, and LIPI categories with irAEs was evaluated by univariate and multivariate logistic regression analyses, providing an odds ratio (OR) with a 95% CI. All variables were included in the final multivariate models depending on their statistical significance in univariate testing. The estimate and comparison of PFS and OS were performed by the Kaplan–Meier method and a two-sided log-rank test, respectively. A multivariate Cox regression model was applied to evaluate the correlation of predefined covariates with survival, providing a hazard ratio (HR) with a 95% CI. All tests were two-sided, and statistical significance was set at a *p* value less than 0.05. SPSS (IBM SPSS Statistics for Windows, version 23.0, Armonk, NY, USA) and Prism (GraphPad Software version 9.0 for Windows, Boston, Massachusetts, USA) software allowed statistical analysis and figure rendering, respectively. R software version 4.1.2 (R Foundation for Statistical Computing, Vienna, Austria) and the library MatchIt were used for PSM [27].

## 3. Results

### 3.1. Patient Characteristics

We enrolled 345 consecutive patients who met the inclusion criteria for this analysis. Between April 2018 and May 2023, 165 patients (47.8%) received first-line pembrolizumab. This was given either as exclusive therapy if their PD-L1 TPS was ≥50% or in combination with platinum-based chemotherapy if their PD-L1 TPS was <50%. From April 2016 to March 2023, 180 patients (52.2%) were treated with PD-(L)1 blockade therapy as second-line treatment after previously receiving platinum-based chemotherapy. In both settings, all participants had metastatic disease extent and ECOG PS ranging from 0 to 2. LIPI categorization at baseline allowed the general population to be divided into three subgroups of similar size. However, we observed a significant imbalance in the distribution of several clinical and pathological characteristics, including ECOG PS, metastatic bone involvement, previous corticosteroid exposure, treatment setting, and first-line chemotherapy agents. After applying PSM, we considered a total of 83 patients relevant for each cohort. This resulted in a homogeneous distribution of all features across the LIPI subgroups. Subsequent evaluations for the predefined endpoints of this research were based on the adjusted PSM populations. Appendix A summarizes the baseline characteristics by LIPI score for both populations.

### 3.2. Changes in LIPI Score

All patients in this study received at least two cycles of treatment involving PD-(L)1 blockade and were evaluated for changes in their LIPI score before the third administration or within 30 days of the second course. Patients with a baseline LIPI score of 0 had a significantly longer duration of anti-PD-(L)1 therapy compared to those with a score of 1 or 2. LIPI-0 patients had a median of 16 treatment cycles, ranging from 2 to 91, while LIPI-1 and LIPI-2 patients had medians of 7 (range 2–19) and 3 (range 2–16) cycles, respectively (*p* < 0.001). During treatment, the distribution of LIPI categories changed notably when reassessing LDH levels and dNLR values. Although LDH levels remained stable, dNLR values decreased significantly across all LIPI groups (Figure 1). As a result, no patients moved into a worse prognosis category, but 20 (24.1%) shifted from LIPI-1 to LIPI-0 and 7 (8.4%) moved from LIPI-2 to LIPI-1 (*p* < 0.001). Multivariate analysis showed that baseline LDH and dNLR levels were not influenced by any clinical or pathological factors. The only factor that had a marginal impact on dNLR values during treatment was pemetrexed-based chemotherapy (Appendix A).

### 3.3. Immune-Related Adverse Events

Over a median observation period of 7.3 (IQR 3.9–15.1) months, we documented a total of 158 irAEs in the PSM population, resulting in an overall incidence rate of 63.5%. Mild to moderate and severe adverse events occurred in 121 (48.6%) and 39 (15.7%) patients, respectively. Immune-related toxicities involved 140 (56.2%) patients, with 18 (7.2%) developing multiple metachronous or synchronous adverse events. Skin toxicities, thyroid dysfunctions, colitis, pneumonitis, and arthritis were the most prevalent immune-related toxicities, with each exceeding a 5% incidence rate among the relevant population. Gastrointestinal and pulmonary adverse events, along with hepatitis and arthritis, were the most commonly reported serious irAEs (Table 1). While we observed no cases of death due to immune-related toxicity, the sequelae of moderate or severe irAEs resulted in treatment discontinuation in a total of 48 (19.2%) patients. On univariate analysis, previous exposure to chest radiotherapy and on-treatment LIPI score were the covariates associated with significantly different frequencies across all grades and severity levels. However, only LIPI classification after the second cycle of therapy was confirmed to be an independent predictor at multivariate analysis, with patients in the good group having significantly higher odds of irAEs than those in the intermediate group. Although changes in dNLR values were associated with a significant difference in all grades and serious adverse events in the univariate comparison, their predictive potential was not confirmed by the multivariate testing. ROC analysis validated the inconsistency of the distributions of LDH levels and dNLR values at both time points in predicting the likelihood of irAEs, regardless of their grading (Appendix A). It is also worth noting that patients who underwent a change in LIPI classification had significantly higher odds of experiencing all grades and severe irAEs, as confirmed by both univariate and multivariate analyses. Table 2 details the logistic regression analysis of immune-related toxicities in the PSM population.

### 3.4. Survival Outcomes

After a median follow-up of 21.1 (95% CI 17.9–25.8) months, 28 (23.1%) patients who were undergoing upfront treatment did not experience any disease progression, while 14 (10.9%) patients on second-line therapy had the same outcome (*p* = 0.010). Additionally, 33 (27.3%) patients who received first-line therapy were censored without any events relevant to survival, in comparison to 17 (13.3%) patients on second-line therapy (*p* = 0.006). On univariate analysis, LIPI categorization at baseline and at the second time point showed a significant impact on survival, with good-risk patients experiencing longer PFS and OS in both treatment settings (Table 3, Figure 2 and Figure 3). Multivariate analysis included clinical and pathological covariates presumably affecting survival. While the LIPI score at baseline maintained its independent effect on both PFS and OS in either the first- or second-line setting, on-treatment LIPI score was only found to be a significant covariate with influence on OS. It is also noteworthy that previous exposure to corticosteroids was consistently identified as an independent prognostic factor for all survival assessments in the multivariable Cox models (Table 4).

## 4. Discussion

In this retrospective analysis, we investigated whether early changes in LIPI score during ICI-based therapy could serve as a predictor for irAEs in advanced NSCLC patients. Initially, the categorization at baseline did not reveal any significant difference in the risk of immune-related toxicities. However, upon reassessment after two treatment cycles, it became evident that patients with a favorable LIPI score had a significantly higher incidence of irAEs across all grades, ranging from mild to severe. This finding held consistent regardless of the specific treatment types and settings. Additionally, the distribution of LIPI groups at both time points had a relevant impact on survival in this patient population. These findings, which have not been previously reported, require a critical appraisal of their clinical relevance and raise several matters for discussion.

The current research is based on a methodological framework that involves retrospectively analyzing real-life data from a single center. This approach implies inherent strengths and weaknesses. Real-world studies have proven to be valuable in cancer research as they bridge the gap between clinical trials and routine practice, providing evidence for various stakeholders [28]. In this study, we used a retrospective analysis methodology to explore the predictive potential of changes in LIPI scoring, which would be difficult to examine through prospective research [29]. Our approach involved conducting a PSM of whole series, adhering to best practice guidelines for medical research [30]. In this regard, we considered all relevant clinical, pathological, and pharmacological covariates for optimal prognostic weighting [31]. The reliability of our medical records is ensured by their close alignment with the government registry for monitoring the reimbursement of high-cost drugs, including all anti-PD-(L)1 agents under investigation.

A first key issue concerns the incidence of irAEs in our series. We found that 65.3% of patients experienced at least one immune-related toxicity of any grade. While the proportion of severe irAEs is consistent with that described in similar real-world studies [11,32], the incidence of mild to moderate toxicities appears significantly higher, closely resembling figures observed in clinical trials [33,34,35]. Several reasons may account for this discrepancy. Patients enrolled in our study were rigorously monitored according to the rules of the regulatory agency. These guidelines require a comprehensive reassessment of the patient’s condition and laboratory tests before prescribing the next course of treatment. All irAEs were recorded in a monitoring registry that adhered to the criteria of good clinical practice. Although the average observation period during treatment was 7.3 months, most patients had a longer follow-up duration of over 21 months. This allowed for a better identification of late-onset irAEs, which have been reported to occur in more than 51% and 57% of cases at 12 and 24 months after treatment initiation, respectively [36]. The fact that this research was conducted in a single center also minimized variability in interpretation among different physicians. These approaches likely led to more reliable detection of asymptomatic or mildly symptomatic irAEs, which are generally overlooked in purely retrospective studies.

A second topic for discussion concerns the reliability of the results regarding the primary endpoints of our research. After conducting a multivariate logistic regression analysis, we found no significant correlation between the initial categorization of LIPI and the risk of developing irAEs. This is consistent with the findings presented by Pierro et al. in their retrospective analysis of elderly patients with different diagnoses who received anti-PD-(L)1 agents [19]. However, it challenges the results of Sonehara et al., who addressed the predictive impact of pre-treatment LIPI groups on irAEs in a retrospective series of patients with advanced NSCLC [17]. In this study, multivariate analyses based on logistic regression revealed a score of 0 or 1 on baseline LIPI as an independent predictive factor that correlated with the development of irAEs. In contrast, our multivariate model did not confirm the predictive value of the LIPI score at baseline but instead identified an early change in LIPI classification as a relevant predictor for the occurrence of immune-related toxicities of any grade. This implies that it is not so much the simple pre-treatment LIPI score as its longitudinal variation that provides predictive insights into the possible development of irAEs. It is worth noting that the changes in LIPI categorization primarily affected patients with intermediate scores, with a significant percentage transitioning to the good group. Most of these patients also experienced irAEs, which could explain the statistical significance of the post-treatment assessment. In addition, only a minority of patients experienced a change in LDH levels. This finding reflects the results of a post hoc analysis of two large randomized trials of atezolizumab as second-line therapy for advanced NSCLC, implying that the switch in LIPI scoring did not rely on variations in LDH [22]. As expected, dichotomization of patients according to LDH level (within or beyond the ULN) did not provide any predictive information on the occurrence of irAEs at any of the time points. The latter result finds opposing views in the available evidence. Several studies have confirmed or excluded the role of LDH testing in the management of immune-related toxicities [37,38,39,40]. In contrast to what was observed for LDH, dNLR changed substantially during treatment, showing a reduction below the threshold value of 3 in both patients who had a favorable and intermediate LIPI score at baseline. Since neutrophils are the most abundant circulating leukocytes, it is reasonable to assume that a decreasing variation in their absolute counts resulted in a significant reduction in dNLR for these subgroups. Our multivariate analysis model showed that only pemetrexed-based chemotherapy exerted a marginally significant effect on dNLR values. Consistent with previous observations, this suggests that the dynamic changes are presumably the result of the PD-(L)1 signaling blockade [20,21,41,42]. In disagreement with the results of Eun et al., our multivariate logistic regression analysis revealed that categorization of patients according to their dNLR value (less than or equal to/above 3) did not independently predict the development of irAEs [43]. This finding was unexpected and suggests that simple processing of bloodstream leukocyte subpopulations is unable to capture immune reactivity following treatment with anti-PD-(L)1 agents. We hypothesized that combining information on dNLR and LDH might more comprehensively reflect tissue damage during treatment and mirror the tumor microenvironment and systemic response to ICI therapy [44]. In this regard, a large retrospective analysis revealed a significant positive correlation between a high tumor mutational burden and irAEs during anti-PD-1 therapy in a wide range of solid malignancies [45].

The last issue involves the secondary endpoints of this study. Multivariate Cox regression analysis confirmed the pre-treatment LIPI score as an independent factor for both PFS and OS. Consistent with several previous observations, patients with a good LIPI score at baseline had significantly better outcomes than the intermediate or poor LIPI groups [46]. As previously reported by Xiong et al. in an advanced NSCLC population who received a PD-1 inhibitor in combination with chemotherapy as upfront treatment [47], we revealed that early post-treatment LIPI scores had the same significant impact on survival outcomes. Continuous assessment of LIPI consistently influenced survival rates, regardless of treatment factors, including the type and specific line of therapy received. These data are relevant in suggesting that the clinical utility of this tool lies in its prognostic ability [48].

Current research recognizes several constraints, including, but not limited to, the following instances. Although the source of our data is a prospective government registry, the design of this study relies on a retrospective analysis. This implies a selection bias that PSM can mitigate but not completely eliminate. In this regard, we included all patients consecutively, excluding only those who had received less than two cycles of treatment. Since early discontinuation is usually motivated by rapid clinical progression, their exclusion may have had a non-negligible impact on the assessment of irAEs and disease outcomes [49]. The prognostic value of LIPI categorization introduces an immortal time bias that was unpredictable at baseline [50]. Prolonged survival indeed results in a longer duration of anti-PD-(L)1 blockade, which consequently increases the risk of immune-related toxicities [51]. Of the total irAEs, nine (5.7%) occurred before the sixth week of treatment, which is the minimum observation period before the third treatment cycle. Although this percentage is small, it is evident that changes in LIPI score evaluated prior to the third cycle can only predict subsequent irAEs longitudinally. Finally, data analysis in an experimental setting not previously investigated required several multivariable comparisons, which may have led to alpha risk inflation. These observations imply an increased likelihood of false-positive results arising from multivariate regression analyses, the significance of which should be considered suggestive of further research hypotheses.

## 5. Conclusions

Our findings highlight the flexibility of LIPI categorization, as evidenced by its early changes on PD-(L)1 blockade in advanced NSCLC. The adjustment of LIPI scores mainly involves patients with intermediate risk through a reduction in their dNLR. Patients who maintain a good LIPI score and those moving from the intermediate to good group would be at higher risk for developing irAEs. This study adds to the literature by suggesting that reassessing the LIPI score after two courses of treatment can serve as a valuable tool in identifying patients who should be closely monitored for potential immune-related toxicities. Our survival analysis also supports the hypothesis that LIPI dynamic changes reflect the prognosis of this population. However, the inherent shortcomings of this study and discordance with available evidence recommend caution in the interpretation of these results and warrant further confirmation in independent series. Whether these findings can be extended to the combination of anti-PD-(L)1 and anti-CTLA-4 agents or to different tumor types requires additional investigation.

## Figures and Tables

**Figure 1 cancers-16-00453-f001:**
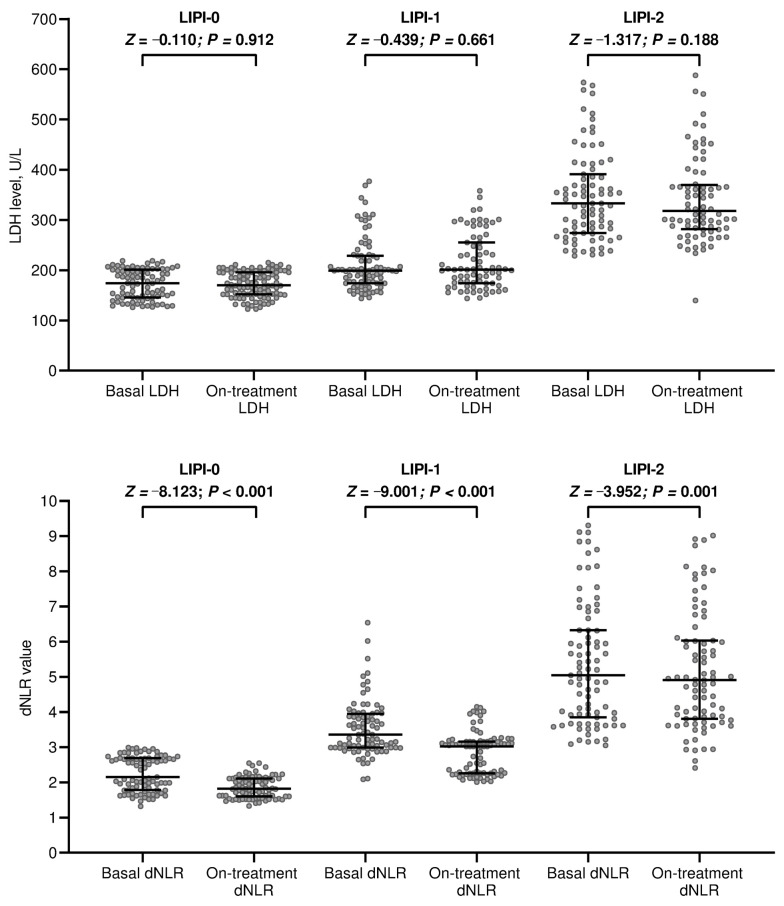
Dynamic changes in LIPI determinants. LIPI, lung immune prognostic index; LDH, lactate dehydrogenase; dNLR, derived neutrophil/lymphocyte ratio. Bars represent median values with interquartile range.

**Figure 2 cancers-16-00453-f002:**
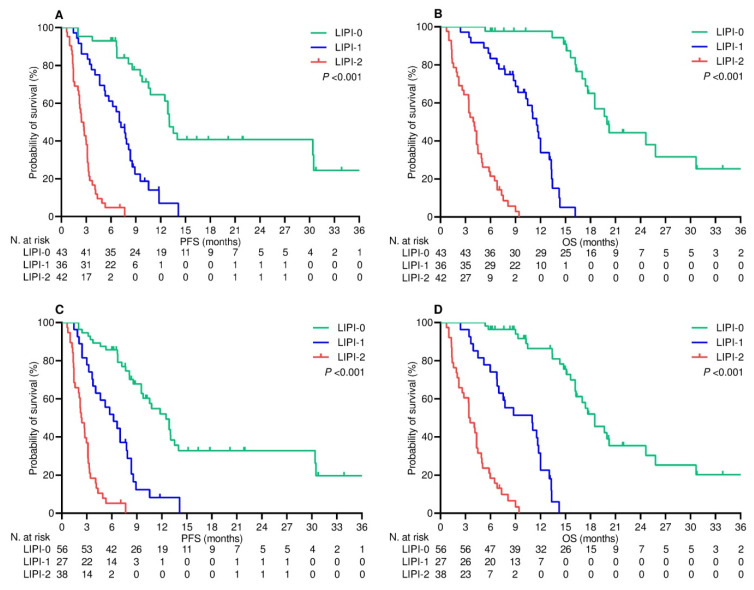
First-line therapy survival outcomes according to LIPI categories. (**A**) progression-free survival depending on LIPI categorization at baseline; (**B**) overall survival depending on LIPI categorization at baseline; (**C**) progression-free survival depending on LIPI categorization after two cycles of treatment; (**D**) overall survival depending on LIPI categorization after two cycles of treatment.

**Figure 3 cancers-16-00453-f003:**
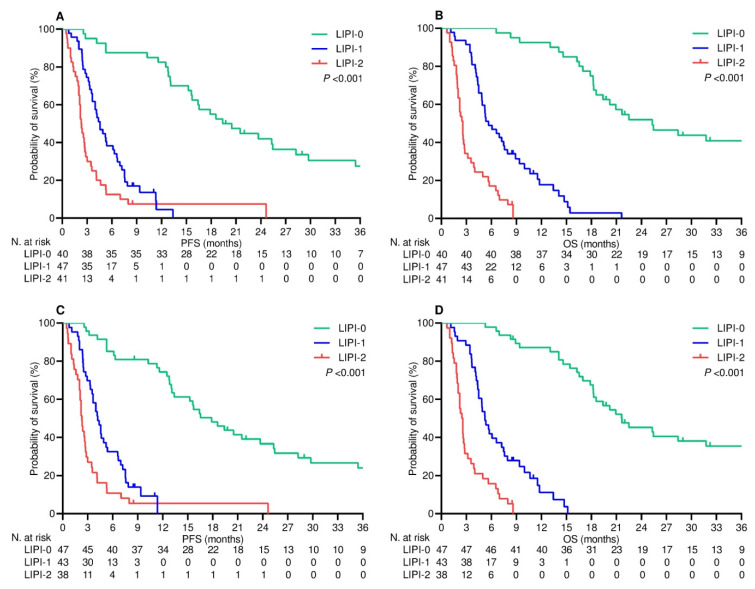
Second-line therapy survival outcomes according to LIPI categories. (**A**) progression-free survival depending on LIPI categorization at baseline; (**B**) overall survival depending on LIPI categorization at baseline; (**C**) progression-free survival depending on LIPI categorization after two cycles of treatment; (**D**) overall survival depending on LIPI categorization after two cycles of treatment.

**Table 1 cancers-16-00453-t001:** Immune-related adverse events in PSM population (N = 249).

irAE Typology	All Grades, No. of Patients (%)	Grade 1–2, No. of Patients (%)	Grade 3–4, No. of Patients (%)	Median Time to Onset, Weeks (IQR)
Dermatologic	30 (12.0%)	27 (10.8%)	3 (1.2%)	7.4 (2.9–16.8)
Thyroid dysfunction				
- Hypothyroidism	24 (9.6%)	23 (9.2%)	1 (0.4%)	10.9 (6.3–20.2)
- Hyperthyroidism	5 (2.0%)	3 (1.2%)	2 (0.8%)	7.9 (6.2–9.4)
Colitis	18 (7.2%)	13 (5.2%)	5 (2.0%)	6.4 (2.5–26.9)
Pneumonitis	17 (6.8%)	11 (4.4%)	6 (2.4%)	13.8 (5.8–19.8)
Hepatitis	13 (5.2%)	8 (3.2%)	5 (2.0%)	5.8 (2.1–18.1)
Arthritis	13 (5.2%)	8 (3.2%)	5 (2.0%)	39.6 (9.2–50.1)
Pancreatitis	12 (4.8%)	9 (3.6%)	3 (1.2%)	12.7 (4.6–21.4)
Myositis	6 (2.4%)	4 (1.6%)	2 (0.8%)	12.7 (5.9–22.6)
Nephritis	5 (2.0%)	3 (1.2%)	2 (0.8%)	11.9 (3.4–20.3
Diabetes	3 (1.2%)	3 (1.2%)	-	11.8 (5.2–18.3)
Hypophysitis	3 (1.2%)	2 (0.8%)	1 (0.4%)	19.9 (6.6–33.6)
Vasculitis	3 (1.2%)	2 (0.8%)	1 (0.4%)	4.1 (3.2–5.9)
Adrenal dysfunction	2 (0.8%)	2 (0.8%)	-	10.3 (8.6–29.1)
Peripheral sensory neuropathy	2 (0.8%)	1 (0.4%)	1 (0.4%)	5.4 (2.5–29.3)
Uveitis	2 (0.8%)	1 (0.4%)	1 (0.4%)	9.5 (7.2–38.2)
Myocarditis	2 (0.8%)	1 (0.4%)	1 (0.4%)	4.8 (4.1–11.3)

PSM, propensity score matching; irAE, immune-related adverse event; IQR, interquartile range.

**Table 2 cancers-16-00453-t002:** Analysis of irAEs.

Covariate	Univariate Analysis		Multivariate Analysis		Univariate Analysis		Multivariate Analysis		Univariate Analysis		Multivariate Analysis	
	Any Grade irAEs	*p* Value	OR (95% CI)	*p* Value	Grade 1–2 irAEs	*p* Value	OR (95% CI)	*p* Value	Grade 3–4 irAEs	*p* Value	OR (95% CI)	*p* Value
Age		0.458	-	-		0.164	-	-		0.331	-	-
- <70 years (N = 110)	67 (60.9%)				48 (46.3%)				20 (18.2%)			
- ≥70 years (N = 139)	91 (65.5%)				73 (52.5%)				19 (48.7%)			
Sex		0.075	-	-		0.014		0.025		0.121	-	-
- Female (N = 76)	42 (55.3%)				28 (36.8%)		1.00		16 (21.1%)			
- Male (N = 173)	116 (67.0%)				93 (53.8%)		1.96 (1.09–3.54)		23 (13.3%)			
ECOG PS		0.306	-	-		0.370	-			0.767	-	-
- 0 or 1 (N = 200)	130 (65.0%)				100 (50.0%)				32 (16.0%)			
- 2 (N = 49)	28 (57.1%)				21 (42.8%)				7 (14.3%)			
Histologic subtype		0.054	-	-		0.087	-			0.430	-	-
- Nonsquamous (N = 185)	111 (60.0%)				84 (45.4%)				27 (14.6%)			
- Squamous (N = 64)	47 (73.4%)				37 (57.8%)				12 (18.8%)			
Number of metastatic sites		0.496	-	-		0.664	-	-		0.970	-	-
- ≤2 (N = 122)	80 (65.6%)				61 (49.2%)				19 (15.6%)			
- >2 (N = 127)	78 (61.4%)				60 (47.2%)				20 (15.7%)			
Bone metastases		0.818	-	-		0.199	-	-		0.393	-	-
- Non present (N = 180)	115 (63.9%)				92 (51.1%)				26 (14.4%)			
- Any (N = 69)	43 (62.3%)				29 (42.0%)				13 (18.8%)			
Brain metastases		0.745	-	-		0.933	-	-		0.003		0.006
- Non present (N = 197)	124 (62.9%)				96 (48.7%)				24 (12.2%)		1.00	
- Any (N = 52)	34 (65.4%)				25 (48.1%)				15 (28.8%)		3.02 (1.36–6.68)	
Liver metastases		0.353	-	-		0.131	-	-		0.939	-	-
- Non present (N = 218)	136 (62.4%)				102 (46.8%)				34 (15.6%)			
- Any (N = 31)	22 (71.0%)				198 (61.3%)				5 (16.1%)			
PD-L1 TPS		0.014				0.289	-	-		0.265	-	-
< 1% (N = 131)	86 (65.6%)		1.00	-	69 (52.7%)				22 (16.8%)			
≥1% and ≤49% (N = 77)	54 (70.1%)		1.15 (0.60–2.20)	0.669	36 (46.7%)				14 (18.2%)			
≥50% (N = 41)	18 (43.9%)		0.41 (0.17–0.96)	0.041	16 (39.0%)				3 (7.3%)			
BMI (kg/m^2^)		0.237	-	-		0.290	-	-		0.357	-	-
- <25 (N = 130)	78 (60.0%)				59 (45.4%)				23 (17.7%)			
- ≥25 (N = 119)	80 (67.2%)				62 (52.1%)				16 (13.4%)			
Smoking habits		0.042	-	0.171		0.315	-	-		0.708	-	-
- Never (N = 30)	14 (46.7%)		1.00		12 (40.0%)				4 (13.3%)			
- Ever (N = 219)	144 (65.7%)		1.81 (0.77–4.24)		109 (49.8%)				35 (20.0%)			
Previous thoracic RT		<0.001	-	0.998		0.006		0.006		0.015		0.161
- No (N = 220)	129 (58.6%)		1.00		100 (45.5%)		1.00		30 (13.6%)		1.00	
- Yes (N = 29)	29 (100%)		NA		21 (72.4%)		3.43 (1.41–8.36)		9 (31.0%)		1.79 (0.76–5.02)	
Autoimmune disease		0.060	-	-		0.085	-	-		0.019		0.080
- No (N = 243)	152 (62.5%)				116 (47.7%)				36 (14.8%)		1.00	
- Yes (N = 6)	6 (100%)				5 (83.3%)				3 (50.0%)		4.61 (0.83–25.60)	
Corticosteroids ^a^ (N = 110)	61 (55.5%)	0.020	0.67 (0.37–1.20)	0.186	50 (45.5%)	0.378	-	-	10 (9.1%)	0.011	0.32 (0.13–0.74)	0.008
APAP (N = 89) ^b^	52 (58.4%)	0.219	-	-	44 (49.4%)	0.843	-	-	12 (13.5%)	0.480	-	-
Systemic antibiotics (N = 54) ^c^	29 (53.7%)	0.093	-	-	21 (38.9%)	0.107	-	-	9 (16.7%)	0.819	-	-
PPI (N = 92)	63 (68.5%)	0.208	-	-	51 (55.4%)	0.098	-	-	12 (13.0%)	0.384	-	-
Statins (N = 86)	54 (62.8%)	0.875	-	-	38 (44.2%)	0.312	-	-	19 (22.1%)	0.043	2.36 (1.11–5.03)	0.026
Fibrates (N = 42)	21 (50.0%)	0.047	0.78 (0.36–1.67)	0.523	17 (40.5%)	0.248	-	-	7 (16.7%)	0.844	-	-
NSAIDs or ASA (N = 54)	40 (74.1%)	0.067	-	-	30 (55.6%)	0.247	-	-	8 (14.8%)	0.846	-	-
Beta-blockers (N = 60)	30 (50.0%)	0.013	0.56 (0.29–1.11)	0.098	19 (31.7%)	0.003	0.40 (0.20–0.77)	0.007	6 (10.0%)	0.166	-	-
ACEi or ARBs (N = 81)	53 (65.4%)	0.653	-	-	42 (51.9%)	0.475	-	-	9 (11.1%)	0.170	-	-
Metformin (N = 72)	45 (62.5%)	0.842	-	-	32 (44.4%)	0.403	-	-	12 (16.7%)	0.781	-	-
Oral or transdermal opioids (N = 106)	69 (65.1%)	0.643	-	-	53 (50.0%)	0.702	-	-	17 (16.0%)	0.888	-	-
Treatment setting		0.074	-	-		0.064	-	-		0.740	-	-
- first line (N = 121)	70 (57.8%)				51 (42.1%)				18 (14.9%)			
- second line (N = 128)	88 (68.7%)				70 (54.7%)				21 (16.4%)			
Treatment type		0.257	-	-		0.984	-	-		0.081	-	-
- ICI only (N = 168)	102 (60.7%)				81 (48.2%)				22 (13.1%)			
- pemetrexed-based (N = 67)	48 (71.6%)				33 (49.2%)				16 (23.9%)			
- paclitaxel-based (N = 14)	8 (57.1%)				7 (50.0%)				1 (7.1%)			
ICI		0.022				0.004				0.787	-	-
- Nivolumab (N = 79)	58 (73.4%)		1.00	-	48 (60.7%)		1.00	-	14 (17.7%)			
- Pembrolizumab (N = 159)	91 (57.2%)		0.62 (0.31–1.22)	0.169	65 (40.9%)		0.46 (0.26–0.83)	0.010	23 (14.4%)			
- Atezolizumab (N = 11)	9 (81.8%)		0.51 (0.33–8.91)	0.512	8 (72.7%)		1.98 (0.45–8.58)	0.358	2 (18.2%)			
Basal LDH level		0.051	-	-		0.443	-	-		0.185	-	-
- ≤ULN (N = 142)	98 (69.0%)				72 (50.7%)				26 (18.3%)			
- >ULN (N = 107)	60 (56.1%)				49 (45.8%)				13 (12.1%)			
Basal dNLR value		0.409	-	-		0.799	-	-		0.135	-	-
- ≤3 (N = 107)	71 (66.3%)				51 (47.7%)				21 (19.6%)			
- >3 (N = 142)	87 (61.2%)				70 (49.3%)				18 (12.7%)			
Basal LIPI score		0.059	-	-		0.894	-	-		0.177	-	-
- 0 (N = 83)	61(73.5%)				42 (50.6%)				18 (21.7%)			
- 1 (N = 83)	47 (56.6%)				39 (47.0%)				11 (13.2%)			
- 2 (N = 83)	50 (60.2%)				40 (48.2%)				10 (12.0%)			
On-treatment LDH level		0.067	-	-		0.597	-	-		0.069	-	-
- ≤ULN (N = 146)	100 (68.5%)				73 (50.0%)				28 (19.2%)			
- >ULN (N = 103)	58 (56.3%)				48 (46.6%)				11 (10.7%)			
On-treatment dNLR value		0.002		0.932		0.331	-	-		0.001		0.144
- ≤3 (N = 130)	94 (72.3%)		1.00		67 (51.5%)				30 (23.1%)		1.00	
- >3 (N = 119)	64 (53.8%)		1.04 (0.37–2.89)		54 (45.4%)				9 (7.6%)		0.36 (0.09–1.41)	
On-treatment LIPI score		0.001				0.028				0.016		
- 0 (N = 103)	79 (76.7%)		1.00	-	58 (56.3%)		1.00	-	23 (22.3%)		1.00	-
- 1 (N = 70)	36 (51.4%)		0.33 (0.13–0.85)	0.023	25 (35.7%)		0.47 (0.25–0.88)	0.020	11 (15.7%)		0.31 (0.11–0.90)	0.031
- 2 (N = 76)	43 (56.5%)		0.47 (0.14–1.58)	0.226	38 (50.0%)		0.90 (0.49–1.68)	0.761	5 (6.5%)		0.43 (0.42–1.62)	0.432
On-treatment LIPI change		0.001		0.012		0.074	-	-		0.001		0.023
- No (N = 222)	133 (59.9%)		1.00		103 (46.4%)				29 (13.0%)		1.00	
- Yes (N = 27)	25 (92.6%)		7.03 (1.54–32.03)		18 (66.7%)				10 (37.0%)		2.84 (1.15–7.02)	

irAEs, immune-related adverse events; OR, odds ratio; CI, confidence interval; ECOG PS, Eastern Cooperative Oncology Group Performance Status; PD-L1 TPS, programmed cell death ligand-1 tumor proportion score; BMI, body mass index; RT, radiotherapy; APAP, acetaminophen; PPI, proton pump inhibitors; NSAIDs, nonsteroidal anti-inflammatory drugs; ASA, acetylsalicylic acid; ACEi, angiotensin-converting enzyme inhibitors; ARBs, angiotensin II type 2 receptor blockers; ICI, immune checkpoint inhibitor; LDH, lactate dehydrogenase; dNLR, derived neutrophil/lymphocyte ratio; LIPI, lung immune prognostic index. ^a^ Corticosteroids indicate intake of prednisone equivalent ≥10 mg daily for at least 5 days before 14 days prior to the start of treatment (excluding premedication for chemotherapy); ^b^ APAP indicates a therapeutic intake of at least 1000 mg per day for more than 24 h during the 30 days prior to the start of treatment; ^c^ systemic antibiotics indicate a therapeutic intake in the 30 days prior to the start of treatment.

**Table 3 cancers-16-00453-t003:** Univariate analysis of survival.

Covariate	First-Line Therapy Population				Second-Line Therapy Population			
	Median PFS (95% CI), Months	*p* Value	Median OS (95% CI), Months	*p* Value	Median PFS (95% CI), Months	*p* Value	Median OS (95% CI), Months	*p* Value
Basal LIPI score		<0.001		<0.001		<0.001		<0.001
- 0 (N = 83)	13.0 (11.9–14.1)		19.9 (17.3–22.6)		19.4 (13.9–24.8)		25.3 (16.6–33.9)	
- 1 (N = 83)	7.0 (5.5–8.4)		11.5 (10.4–12.6)		4.5 (3.5–5.5)		5.6 (3.8–7.5)	
- 2 (N = 83)	2.4 (1.8–2.9)		3.9 (3.3–4.5)		2.3 (1.9–2.6)		2.5 (2.1–3.0)	
On-treatment LIPI score		<0.001		<0.001		<0.001		<0.001
- 0 (N = 103)	12.5 (9.9–15.0)		18.5 (15.5–21.4)		17.8 (13.6–22.0)		21.6 (15.4–27.8)	
- 1 (N = 70)	6.2 (3.7–8.6)		11.0 (5.6–16.3		4.2 (3.3–5.0)		5.2 (4.2–6.2)	
- 2 (N = 76)	2.3 (1.5–3.0)		3.4 (2.5–4.2)		2.2 (1.9–2.4)		2.4 (2.0–2.9)	

PFS, progression-free survival; OS, overall survival; HR, hazard ratio; CI, confidence interval; LIPI, lung immune prognostic index.

**Table 4 cancers-16-00453-t004:** Multivariate analysis of survival.

Covariate	First-Line Therapy Population				Second-Line Therapy Population			
	Progression-Free Survival		Overall Survival		Progression-Free Survival		Overall Survival	
	HR (95% CI)	*p* Value	HR (95% CI)	*p* Value	HR (95% CI)	*p* Value	HR (95% CI)	*p* Value
Age		0.770		0.763		0.885		0.404
- <70 years	1.00		1.00		1.00		1.00	
- ≥70 years	0.928 (0.56–1.56)		0.92 (0.56–1.52)		1.07 (0.63–1.68)		0.81 (0.51–1.31)	
Sex		0.475		0.399		0.187		0.296
- Female	1.00		1.00		1.00		1.00	
- Male	0.824 (0.48–1.40)		0.78 (0.45–1.37)		0.72 (0.45–1.16)		0.77 (0.47–1.25)	
ECOG PS		0.493		0.856		0.742		0.484
- 0 or 1	1.00		1.00		1.00		1.00	
- 2	1.24 (0.66–2.31)		0.94 (0.49–1.80)		1.10 (0.62–1.94)		1.22 (0.69–2.16)	
Histologic subtype		0.050		0.150		0.952		0.685
- Nonsquamous	1.00		1.00		1.00		1.00	
- Squamous	0.47 (0.22–1.00)		0.57 (0.27–1.22)		0.98 (0.62–1.55)		0.91 (0.58–1.41)	
Number of metastatic sites		0.020		0.001		0.138		0.080
- ≤2	1.00		1.00		1.00		1.00	
- >2	2.08 (1.12–3.86)		3.14 (1.61–6.11)		1.63 (0.85–3.11)		1.81 (0.93–3.54)	
Bone metastases		0.714		0.999		0.856		0.344
- Not present	1.00		1.00		1.00		1.00	
- Any	0.89 (0.51–1.58)		1.01 (0.56–1.76)		1.05 (0.59–1.87)		0.73 (0.39–1.38)	
Brain metastases		0.042		0.002		0.154		0.734
- Not present	1.00		1.00		1.00		1.00	
- Any	0.53 (0.29–0.97)		0.37 (0.20–0.68)		1.58 (0.84–3.00)		0.90 (0.50–1.62)	
Liver metastases		0.238		0.745		0.304		0.023
- Not present	1.00		1.00		1.00		1.00	
- Any	1.56 (0.74–3.26)		0.87 (0.39–1.95)		0.70 (0.35–1.37)		0.44 (0.22–0.89)	
PD-L1 TPS								
- <1%	1.00	0.028	1.00	0.002	1.00	0.522	1.00	0.829
- ≥1% and ≤49%	1.92 (0.94–3.92)	0.072	1.68 (0.82–3.45)	0.152	1.17 (0.75–1.81)	0.482	0.89 (0.56–1.41)	0.624
- ≥50%	2.17 (1.14–4.12)	0.018	3.59 (1.77–7.28)	0.001	0.75 (0.32–1.77)	0.522	0.80 (0.33–1.92)	0.619
BMI (kg/m^2^)		0.547		0.651		0.815		0.209
- <25	1.00		1.00		1.00		1.00	
- ≥25	1.16 (0.70–1.93)		0.88 (0.51–1.50)		1.05 (0.68–1.62)		0.75 (0.48–1.17)	
Corticosteroids ^a^		0.012		0.001		0.010		0.046
- No	1.00		1.00		1.00		1.00	
- Yes	1.89 (1.14–3.12)		2.53 (1.49–4.28)		1.77 (1.14–2.74)		1.56 (1.01–2.39)	
APAP ^b^		0.010		0.197		0.786		0.743
- No	1.00		1.00		1.00		1.00	
- Yes	2.24 (1.21–4.13)		1.52 (0.80–2.87)		1.06 (0.96–1.62)		1.07 (0.69–1.67)	
Basal LIPI score								
- 0	1.00	<0.001	1.00	<0.001	1.00	0.012	1.00	0.001
- 1	4.22 (1.73–>10)	0.001	>10 (5.1–>10)	0.001	5.08 (1.74–>10)	0.003	7.36 (2.57–>10)	<0.001
- 2	19.95 (4.00–>10)	<0.001	>100 (NA)	<0.001	5.36 (1.08–>10)	0.041	>10 (2.31–>10)	0.005
On-treatment LIPI score								
- 0	1.00	0.514	1.00	0.002	1.00	0.446	1.00	0.041
- 1	1.63 (0.68–3.87)	0.268	3.24 (1.14–9.18)	0.027	1.61 (0.63–4.10)	0.313	3.03 (1.05–8.69)	0.039
- 2	1.98 (0.44–8.79)	0.369	4.12 (1.02–>10)	0.049	3.05 (0.48–>10)	0.237	5.64 (1.02–>10)	0.048

HR, hazard ratio; CI, confidence interval; ECOG PS, Eastern Cooperative Oncology Group Performance Status; PD-L1 TPS, programmed cell death ligand-1 tumor proportion score; BMI, body mass index; APAP, acetaminophen; LIPI, lung immune prognostic index. ^a^ Corticosteroids indicate intake of prednisone equivalent ≥10 mg daily for at least 5 days before 14 days prior to the start of treatment (excluding premedication for chemotherapy); ^b^ APAP indicates a therapeutic intake of at least 1000 mg per day for more than 24 h during the 30 days prior to the start of treatment.

## Data Availability

The datasets generated and analyzed during the current study are available from the corresponding author on reasonable request.

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
