# Peer review of "Early Changes in LIPI Score Predict Immune-Related Adverse Events: A Propensity Score Matched Analysis in Advanced Non-Small Cell Lung Cancer Patients on Immune Checkpoint Blockade"

_cancers, 2024, doi:10.3390/cancers16020453_

Round 1

Reviewer 1 Report

Comments and Suggestions for Authors

The current study investigated the Lung Immune Prognostic Index (LIPI) as a potential biomarker for immune-related adverse events (irAEs) and survival outcomes in advanced non-small cell lung cancer (NSCLC) patients receiving PD-(L)1 blockade-based therapies. The LIPI score, based on neutrophil/lymphocyte ratio and lactate dehydrogenase values, was assessed at baseline and after two treatment cycles. Results showed that changes in LIPI score were associated with irAEs, with the LIPI-0 group having a higher risk of irAEs. LIPI categorization was also linked to overall survival, suggesting LIPI assessment could help identify patients prone to immune-related toxicities.

The retrospective design implies a selection bias that propensity score matching (PSM) can mitigate but not eliminate. Additionally, the exclusion of patients who received less than two cycles of treatment due to rapid clinical progression could have impacted the assessment of irAEs and disease outcomes. Furthermore, the study's findings regarding the predictive value of LIPI score changes for irAEs are somewhat contradictory to other studies. While this study found no significant correlation between initial LIPI categorization and the risk of developing irAEs, other research has shown varying results. This discrepancy warrants further investigation and validation in different settings​​.

Specific comments:

-          The manuscript is overloaded with Tables, some should go to supplemental

-          Should discuss https://pubmed.ncbi.nlm.nih.gov/35140520/ and emphasize what is the novel insight that the current article adds (e.g… longitudinal aspects).

-          The study found a high incidence of irAEs, with mild to moderate toxicities appearing significantly higher than in similar real-world studies. Please explain the cause of this discrepancy. Authors can collect concise information regarding the matter from reviews https://pubmed.ncbi.nlm.nih.gov/36769093/ and https://pubmed.ncbi.nlm.nih.gov/33865201/

-          The study notes that dichotomization of patients based on lactate dehydrogenase (LDH) levels did not provide predictive information on the occurrence of irAEs. In contrast, derived neutrophil/lymphocyte ratio (dNLR) changed significantly during treatment, but this change did not independently predict the development of irAEs.

-          In the caption of Figure 2, "Fist-line therapy survival outcomes according to LIPI categories" should be corrected to "First-line therapy survival outcomes according to LIPI categories"​​.

-          Similarly, in the caption of Figure 3, "Second-line therapy survival outcomes according to LIPI categories" is missing the "r" in "progression-free"​​.

Author Response

Answers to Comments and Suggestions for Authors

The current study investigated the Lung Immune Prognostic Index (LIPI) as a potential biomarker for immune-related adverse events (irAEs) and survival outcomes in advanced non-small cell lung cancer (NSCLC) patients receiving PD-(L)1 blockade-based therapies. The LIPI score, based on neutrophil/lymphocyte ratio and lactate dehydrogenase values, was assessed at baseline and after two treatment cycles. Results showed that changes in LIPI score were associated with irAEs, with the LIPI-0 group having a higher risk of irAEs. LIPI categorization was also linked to overall survival, suggesting LIPI assessment could help identify patients prone to immune-related toxicities.

The retrospective design implies a selection bias that propensity score matching (PSM) can mitigate but not eliminate. Additionally, the exclusion of patients who received less than two cycles of treatment due to rapid clinical progression could have impacted the assessment of irAEs and disease outcomes. Furthermore, the study's findings regarding the predictive value of LIPI score changes for irAEs are somewhat contradictory to other studies. While this study found no significant correlation between initial LIPI categorization and the risk of developing irAEs, other research has shown varying results. This discrepancy warrants further investigation and validation in different settings​​.

Specific comments:

-          The manuscript is overloaded with Tables, some should go to supplemental

- Thank you for these insightful and enlightening comments. Each table given in the main text describes an analytical passage in this study. Although none of them appear redundant, the Reviewer is right in stating that their current number could overload the manuscript. Following this advice, we have moved the more descriptive table (Table 1) to the supplementary material. The numbering of the tables has been changed accordingly.

-          Should discuss https://pubmed.ncbi.nlm.nih.gov/35140520/ and emphasize what is the novel insight that the current article adds (e.g… longitudinal aspects).

- According to the Reviewer's suggestions, we discussed in more detail the results of the study by Sonehara et al. (https://pubmed.ncbi.nlm.nih.gov/35140520). Our findings are not entirely consistent with those of the Japanese Authors, who addressed the predictive impact  of pre-treatment LIPI groups on irAEs in a retrospective series of patients with advanced NSCLC. In this study, multivariate analyses based on logistic regression revealed a score of 0 or 1 on baseline LIPI as an independent predictive factors that correlated with the development of irAEs. In contrast, our multivariate model did not confirm the predictive value of the LIPI score at baseline, but instead identified an early change in LIPI classification as a relevant predictor for the occurrence of immune-related toxicities of any grade. This implies that it is not so much the simple pre-treatment LIPI score as its longitudinal variation that provides predictive insights into the possible development of irAEs.

-          The study found a high incidence of irAEs, with mild to moderate toxicities appearing significantly higher than in similar real-world studies. Please explain the cause of this discrepancy. Authors can collect concise information regarding the matter from reviews https://pubmed.ncbi.nlm.nih.gov/36769093/ and https://pubmed.ncbi.nlm.nih.gov/33865201/

- The discrepancy found in the incidence of mild to moderate irAEs compared with other real-world studies is the first issue we addressed in our discussion. The reasons that might explain the higher prevalence of immune-related toxicities in our series have already been partly explained. In addition, we emphasized the role of regulatory agency rules. These guidelines require a comprehensive reassessment of the patient's condition and laboratory tests before prescribing the next course of treatment. Furthermore, although the average observation period during treatment was 7.3 months, most patients had a longer follow-up duration of over 21 months. This allowed for a better identification of late-onset irAEs, which have been reported to occur in more than 51% and 57% of cases at 12 and 24 months after treatment initiation, respectively (https://pubmed.ncbi.nlm.nih.gov/33865201). Both reviews suggested by the Reviewer were quoted as additional references.

-          The study notes that dichotomization of patients based on lactate dehydrogenase (LDH) levels did not provide predictive information on the occurrence of irAEs. In contrast, derived neutrophil/lymphocyte ratio (dNLR) changed significantly during treatment, but this change did not independently predict the development of irAEs.

- The existing evidence led us to expect that LDH levels at baseline or their changes during treatment would not accurately predict the occurrence of irAEs. However, we were surprised to find that the predictive ability of dNLR ratio values and their variation during treatment was inconsistent in relation to the same outcome. While dNLR values did decrease significantly in response to ICI treatment, their dynamic variation did not emerge as an independent factor for irAEs from multivariate analysis. Our argument is that it is not solely the fluctuation in neutrophil count but rather the combination of dNLR and LDH levels in LIPI categorization that has a predictive impact on the occurrence of immune-related toxicities. This unique finding, which has not been previously reported, serves as the primary discovery of our research..

-          In the caption of Figure 2, "Fist-line therapy survival outcomes according to LIPI categories" should be corrected to "First-line therapy survival outcomes according to LIPI categories"​​.

- This typo has been corrected in the revised version.

 -          Similarly, in the caption of Figure 3, "Second-line therapy survival outcomes according to LIPI categories" is missing the "r" in "progression-free"​​.

- This typo has been corrected in the revised version.

Reviewer 2 Report

Comments and Suggestions for Authors

Excellent paper, reporting interesting information in a comprehensive way. Some terms should be minimally further explained, such as PD-L1 TPS ( line 190-191), for non -oncologist readers.

Graphs and tables are adequate and contribute to a better understanding of the results.

Author Response

Answers to Comments and Suggestions for Authors

Excellent paper, reporting interesting information in a comprehensive way. Some terms should be minimally further explained, such as PD-L1 TPS ( line 190-191), for non -oncologist readers.

Graphs and tables are adequate and contribute to a better understanding of the results.

- We clarified the meaning of PD-L1 TPS by providing a specific description of this term and related procedures that were actually missing in the text. The 22C3 pharmDx anti-PD-L1 antibody and the Dako Omnis platform were used to perform an immunohistochemical assessment of PD-L1 TPS according to the manufacturer's recommendations (Agilent Technologies, Inc.). The TPS was defined as the percentage of at least 100 viable tumor cells exhibiting positive membrane staining for PD-L1 expression.

Reviewer 3 Report

Comments and Suggestions for Authors

Thank you for the giving me a valuable opportunity to review this article.This article is well written about LIPI score to predict outcomes after the PD(L)-1 inhibitors treatment.

There needs minor revisions to publish.

<Materials and methods>

1: Did the patients who developed irAE during the first 2 cycles exclude? If so, please indicate that.   

2: Please show the value of upper limit of LDH level which was used in this research.

3: Authors described “Patients enrolled in our study were rigorously monitored according to the rules of the regulatory agency” in discussion. Please show the “rigorous monitoring rules” in Materials and methods briefly.

 ï¼œConclusions>

4: “Conclusions” is too long. Please summarize your suggestion more simply.

Author Response

Reviewer 3

Answers to Comments and Suggestions for Authors

Thank you for the giving me a valuable opportunity to review this article. This article is well written about LIPI score to predict outcomes after the PD(L)-1 inhibitors treatment.

There needs minor revisions to publish.

<Materials and methods>

1: Did the patients who developed irAE during the first 2 cycles exclude? If so, please indicate that.  

- Patients who experienced any immune-related toxicities even before their third treatment cycle were included in both the univariate and multivariate analysis. The median time to onset of these toxicities, as indicated in the final column of Table 2, was reported in weeks with an interquartile range (IQR). Of the total irAEs, nine (5.7%) occurred before the sixth week of treatment, which is the minimum observation period before the third treatment cycle. Although this percentage is small, it is evident that changes in LIPI score evaluated prior to the third cycle can only predict subsequent irAEs longitudinally. The Reviewer's comment was extremely valuable as it helped us identify this limitation, which we acknowledged in the "Discussion" section of our study.

2: Please show the value of upper limit of LDH level which was used in this research.

- The range of normal variability of LDH was 135 to 225 units per liter (U/L).  As suggested, we have added this information to the subsection "Data Collection and assessments"

3: Authors described “Patients enrolled in our study were rigorously monitored according to the rules of the regulatory agency” in discussion. Please show the “rigorous monitoring rules” in Materials and methods briefly.

- The monitoring agency guidelines require a comprehensive reassessment of clinical conditions as a prerequisite for continued treatment with anti-PD-(L)1 agents. Before each cycle of therapy, all patients underwent a complete physical examination and laboratory tests, including indices of systemic inflammatory response and thyroid, renal, hepatobiliary, pancreatic, adrenocortical, muscle, and pituitary function. As requested by the Reviewer, we have added this brief description in the "Materials and Methods" section.

 ï¼œConclusions>

4: “Conclusions” is too long. Please summarize your suggestion more simply.

- Some statements in the "Conclusion" section actually appeared redundant. Following the Reviewer's suggestion, we have consistently reduced the length of the paragraph. This change should make the concluding remarks easier to understand.

Round 2

Reviewer 1 Report

Comments and Suggestions for Authors

The Authors assessed all raised issues and answered the Reviewer's questions.